https://doi.org/10.1038/s41467-019-08428-2　　**OPEN**

# Modular Design of Programmable Mechanofluorescent DNA Hydrogels

Remi Merindol [1,2,3,6], Giovanne Delechiave[4], Laura Heinen[1,2,3], Luiz Henrique Catalani[4] & Andreas Walther [1,2,3,5]

Mechanosensing systems are ubiquitous in nature and control many functions from cell spreading to wound healing. Biologic systems typically rely on supramolecular transformations and secondary reporter systems to sense weak forces. By contrast, synthetic mechanosensitive materials often use covalent transformations of chromophores, serving both as force sensor and reporter, which hinders orthogonal engineering of their sensitivity, response and modularity. Here, we introduce FRET-based, rationally tunable DNA tension probes into macroscopic 3D all-DNA hydrogels to prepare mechanofluorescent materials with programmable sacrificial bonds and stress relaxation. This design addresses current limitations of mechanochromic system by offering spatiotemporal resolution, as well as quantitative and modular force sensing in soft hydrogels. The programmable force probe design further grants temporal control over the recovery of the mechanofluorescence during stress relaxation, enabling reversible and irreversible strain sensing. We show proof-of-concept applications to study strain fields in composites and to visualize freezing-induced strain patterns in homogeneous hydrogels.

[1] Institute for Macromolecular Chemistry, University of Freiburg, Stefan-Meier-Str. 31, 79104 Freiburg, Germany. [2] Freiburg Materials Research Center, University of Freiburg, Stefan-Meier-Str. 21, 79104 Freiburg, Germany. [3] Freiburg Center for Interactive Materials and Bioinspired Technologies, University of Freiburg, Georges-Köhler-Allee 105, 79110 Freiburg, Germany. [4] Institute of Chemistry, University of São Paulo, 05508-000 São Paulo, Brazil. [5] Freiburg Institute for Advanced Studies (FRIAS), University of Freiburg, Albertstraße 19, 79104 Freiburg, Germany. [6] Present address: Centre De Recherche Paul Pascal, University of Bordeaux, 115 Avenue du Dr Albert Schweitzer, 33600 Pessac, France. Correspondence and requests for materials should be addressed to A.W. (email: Andreas.Walther@makro.uni-freiburg.de)

Living organisms feature complex mechanosensitive systems to sense and adapt to mechanical forces of the environment[1]. Mechanotransduction of mechanical forces into signals controls cell spreading and stem cell differentiation[2,3], and enables wound healing[4] or hearing[5,6] as macroscopic functions. Each function relies on its own mechanosensitive system with unique sensitivity and temporal response and usually combines two main functional components: a force-sensing module and a dynamic structure (actin network, lipid bilayer etc) providing the response. The force-sensing module consists of a supramolecular unit (e.g., folded protein or membrane protein), which undergoes structural changes upon mechanical actuation[7,8]. For instance, the unfolding of the Willbrand glycoprotein reveals cryptic sites that trigger blood clot formation[4]. In biological mechanosensitive systems, the key to performance lies in the cooperative optimization of the sensing mechanism, the processing of the signal through reaction networks, and appropriate support structures triggering the response[9].

In synthetic systems, mechanically induced transformations are known since the discovery of macromolecules; yet, an exploitation of these transformations to design synthetic mechanosensitive materials with custom properties has only emerged relatively recently. Mechanochromic materials, whose optical properties change with mechanical actuation, attract most attention[10–12]. Such materials find use in security applications, for monitoring structural integrity[13], or as sensors to explore complex mechanical processes (e.g., cell/matrix interactions and composite deformation)[14,15]. Early mechanosensitive materials were based on changes in the absorption properties in conjugated polymers[16,17], chromophore aggregates[18,19], or via ring opening of mechanochromic units such as spiropyrans[12,20,21]. Advanced detection, however, requires the development of luminescence/fluorescence signals, which can be detected with a higher spatiotemporal resolution and at a lower concentration[22,23]. Critically, approaching the sensitivity and reversibility of biological systems requires the designing of soft mechanosensitive materials that are able to undergo larger and reversible transformations under weak forces[24]. Yet, despite the progress in such systems[25–28], at present, we lack straight-forward modular and programmable toolboxes that allow us to design macroscopic materials that sense weak forces with a high spatial resolution.

In most current mechanochromic materials, the chromophore serves as both a force sensor and a signaling molecule, which limits the possibilities to independently modulate the sensitivity and reversibility of the system without extensive redesign. FRET-based detection (Forster Resonance Energy Transfer) provides one approach to decouple both processes, as it only takes place when two chromophores are in very close proximity. FRET is routinely used to study cellular function, self-assembly or single-molecule mechanics[29–31]. Molecular tension probes combining DNA structures and FRET pairs have recently emerged to probe into biointerface processes on surfaces or enzyme binding[14,15,32–35]. Yet, these tools have been restricted to 2D interfaces or solutions in which material demand and characterization are simplified. Implementing FRET-based molecular tension probes within macroscopic DNA materials[36–39] would provide a platform for the modular design of mechanosensitive materials operational in 3D, with unmatched modularity, spatiotemporal resolution, and programmable sensitivity.

Here, we realize the integration of FRET-based, rationally tunable DNA tension probes into macroscopic 3D DNA hydrogels to prepare mechanofluorescent materials with programmable sacrificial bonds and stress-relaxation behavior. We demonstrate the assembly of highly stretchable all-DNA hydrogels ( > 500% elongation at break), facile hybridization-driven functionalization with modular force sensors, and quantifiable mechanofluorescent

response. The 3D DNA support structure consists of thermoreversible DNA hydrogels, synthesized enzymatically, which allow shaping and recycling when heated above the melting temperature. The mechanochromic sensors are based on FRET pairs for fluorescent readouts that are maintained in close proximity by weak DNA duplexes that form different sacrificial bond patterns. We use the modularity of our approach to demonstrate reversible and irreversible strain sensing. Finally, we showcase proof-of-concept applications for strain mapping in composites and localization of freezing-induced strain patterns.

## Results

**Design of mechanofluorescent 3D DNA hydrogels**. The assembly and operational principle of the mechanofluorescent DNA hydrogel are presented in Fig. 1. A key aspect in the development of this mechanosensitive 3D DNA system lies in the assembly of a hydrogel matrix, which is easily functionalizable, mechanically stable, and provides access towards macroscopic DNA hydrogels. We used rolling circle amplification (RCA), an isothermal enzymatic process that takes a small circular oligonucleotide template as input and produces a complementary "multiblock" ssDNA that is several thousand nucleotides (nts) in length (Supplementary Figure 1). Recently, we reported how to maximize the yields and suppress the in situ crystallization of $Mg_2P_2O_7$ to facilitate the milligram-scale production of ready-to-use sequence-controlled ssDNA[40]. To form the hydrogels, we designed two 72-nt long templates that yield two "multiblock" hydrogel precursors after RCA. The repeat sequences of each hydrogel precursor present a complementary crosslinking domain between both (12 nts; X/X*) with a melting temperature, $T_m$, of ca. 64 °C (Tables 1 and 2). Additionally, they contain individual barcode domains ($A_x$, $B_x$; $x = 1$, 2, 3, 4; not complementary, 50 nts) to allow for functionalization with the force-sensing module. Upon mixing the two precursors, the crosslinking domains (X/X*) hybridize via supramolecular duplexes linking the hydrogel together (Fig. 1b). Each of the two precursors is synthesized separately, and the hydrogel is formed by annealing a stoichiometric mixture of both.

The force sensor modules are assembled separately from commercial polyacrylamide gel electrophoresis-purified ssDNA via hybridization in a temperature ramp. Figure 1b depicts the general design of the force-sensing modules. They consist of two ssDNAs with a fluorophore (yellow–red dot) and a quencher (red–black dot) that are functionalized at one part to hybridize with the force sensor module (M/M*, N/N*) and at the other end with the complementary sequences ($A_4$*, $B_1$*) of the barcode domains ($A_4$, $B_1$) of the hydrogel. Fluorophore and quencher are brought into close proximity within the force-sensing module (pink), the other strands of which can also hybridize to the barcode domains of the hydrogel precursors. The force-sensing modules can be either connected into a hairpin (HP) or formed via two strands ($D_1/D_1$*; $D_2/D_2$*; further details below). Critically, the design of this module allows to manipulate the force response and can be designed through its sequence and length to be weaker or stronger than the crosslinking domains that hold the network together ($-\Delta G_{D1/D1*} = 46\ kcal\,mol^{-1} > -\Delta G_{X/X*} = 25\ kcal\,mol^{-1} > -\Delta G_{D2/D2*} = 16\ kcal\,mol^{-1}$). After formation, the assembled force-sensing modules are added to the DNA hydrogel, to which they bind isothermally via diffusion onto the barcode domains by the four complementary anchor strands in a two-by-two fashion ($A_x/A_x$*, $B_x/B_x$*; $x = 1$, 4), thereby linking both hydrogel-forming "multiblock" ssDNAs more strongly together. This isothermal hybridization leads to a slight swelling of the hydrogels, but ensures that the correctly folded modules persist during the functionalization and cannot be compromised, even though if unexpected, during heating ramps. At rest, the FRET from

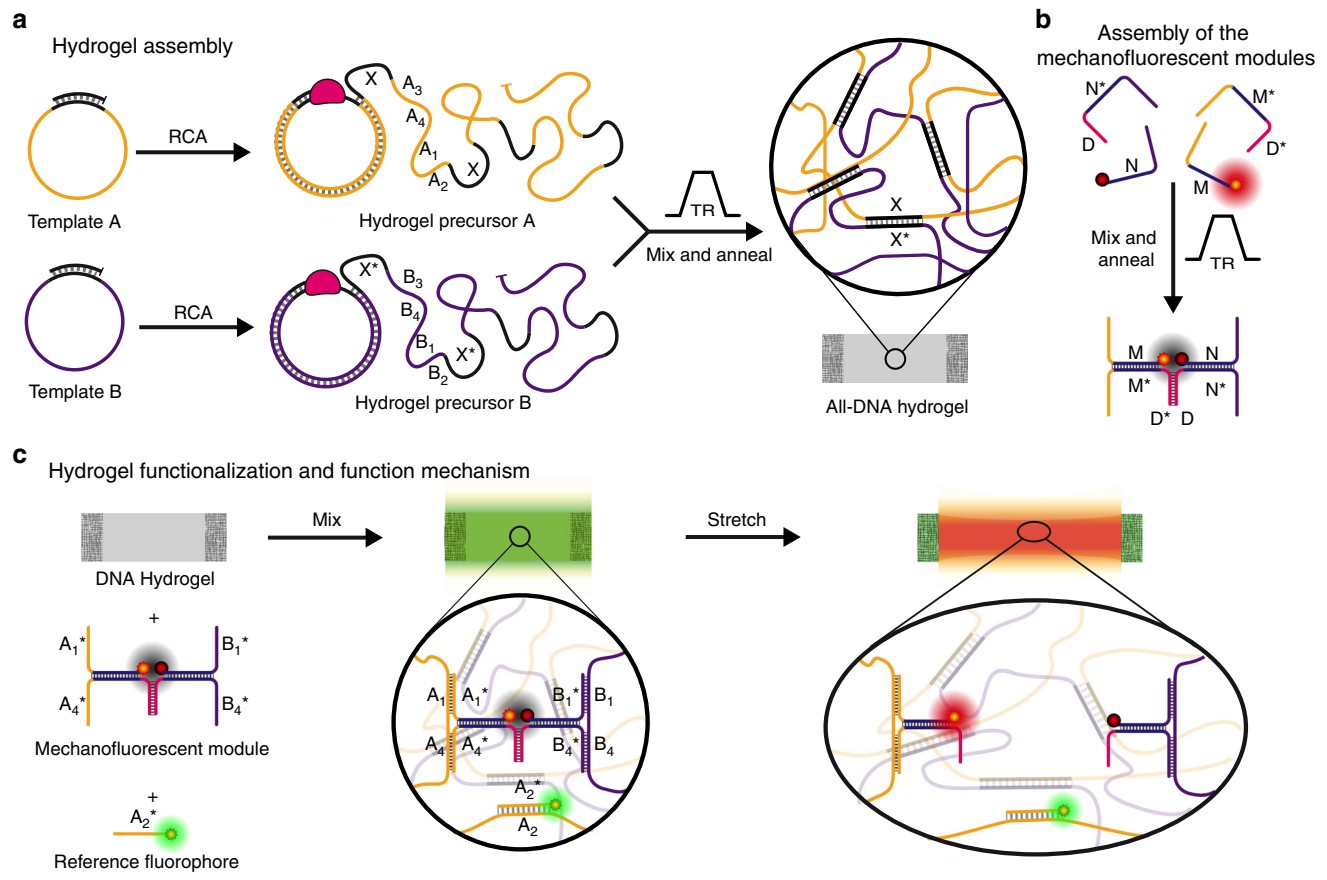

**Fig. 1** Preparation and operational principle of modular mechanofluorescent DNA hydrogels. **a** Enzymatic synthesis of the ssDNA hydrogel precursors and assembly of the DNA hydrogel via X/X* duplex crosslinks. **b** Self-assembly of a typical mechanofluorescent force-sensing module. The red-emitting fluorophore and the quencher are brought in proximity and FRET prevents the emission of red fluorescence. **c** The pristine DNA hydrogel is functionalized with a force-sensing module and with a green-emitting reference fluorophore to form the mechanofluorescent hydrogel. At rest, the hydrogel fluoresces in green as only the reference fluorophore emits light due to an efficient FRET-based quenching in the force-sensing module. Stretching of the hydrogel breaks the sacrificial duplex of the mechanosensing module (in pink), which separates the red-emitting fluorophore from the quencher and decreases FRET. The red-emitting fluorophore starts emitting and the red fluorescence increases compared to the unaffected green fluorescence of the reference. FRET, forster resonance energy transfer

### Table 1 Overview of the DNA sequences and their functions and domains

| Function | Name | Sequence 5′→3′ |
|---|---|---|
| Hydrogel matrix | Hydrogel precursor A ($A_1A_2XA_3A_4$) | (GGTGGCGGCTGACTG GTCAATGAATCGCGT CCGACGTTGACG CTGGATGTAGGATGC GGCGTGTCCACCTAC)~300 |
| | Hydrogel precursor B ($B_1B_2X^*B_3B_4$) | (GCGAAGCGCCCGCTG CCTGTTGAGCGTATC CGTCAACGTCGG GCTGTACCGTTATTG CTCGCGCGGCAGCTC) ~300 |
| Common fluorescent strands | Fluorescent arm $Atto_{565}$-$MA_4^*$ | $Atto_{565}$-CGCGTTGCGCCTGCC GTAGGTGGACACGCC |
| | Quencher arm $B_1^*N$-IowaRQ | CAGCGGGCGCTTCGC GAGCCGCGCACGCCG-IowaRQ |
| | Reference $Atto_{488}$-$A_2^*$ | $Atto_{488}$-ACGCGATTCATTGAC |
| Sensor $D_1$ | Arm $D_1^*$ ($A_1^*M^*D_1^*$) | CAGTCAGCCGCCACC GGCAGGCGCAACGCG CGGCCGCGCGCCCGG |
| | Arm $D_1$ ($D_1N^*B_4^*$) | CCGGGCGCGCGGCCG CGGCGTGCGCGGCTC GAGCTGCCGCGCGAG |
| Sensor $D_2$ | Arm $D_2^*$ ($A_1^*M^*D_2^*$) | CAGTCAGCCGCCACC GGCAGGCGCAACGCG CGTCCGAC |
| | Arm $D_2$ ($D_2N^*B_4^*$) | GTCGGACG CGGCGTGCGCGGCTC GAGCTGCCGCGCGAG |
| Sensor HP | Arm HP ($A_1^*M^*HPN^*B_4^*$) | CAGTCAGCCGCCACC GGCAGGCGCAACGCG CGTCCGACTTTTTTGTCGGACG CGGCGTGCGCGGCTC GAGCTGCCGCGCGAG |
| Control T | Arm T ($A_1^*M^*TN^*BB_4^*$) | CAGTCAGCCGCCACC GGCAGGCGCAACGCG TTTTTT CGGCGTGCGCGGCTC GAGCTGCCGCGCGAG |

**Table 2 Thermodynamic data of relevant duplex motifs**

| Name (abbreviation) | Sequence 5′→3′/Complementary sequence 5′→3′ | $T_m$ (°C)[a] | $-\Delta G$ (kcal mol$^{-1}$)[b] |
|---|---|---|---|
| Cross-linking domain (X/X*) | CCGACGTTGACG/CGTCAACGTCGG | 64 | 25 |
| Barcode module ($A_1$/$A_1$*) | GGTGGCGGCTGACTG/CAGTCAGCCGCCACC | 73 | 32 |
| Barcode module ($A_4$/$A_4$*) | GGCGTGTCCACCTAC/GTAGGTGGACACGCC | 69 | 29 |
| Barcode module ($B_1$/$B_1$*) | GCGAAGCGCCCGCTG/CAGCGGGCGCTTCGC | 76 | 36 |
| Barcode module ($B_4$/$B_4$*) | CTCGCGCGGCAGCTC/GAGCTGCCGCGCGAG | 77 | 38 |
| Barcode reference ($A_2$/$A_2$*) | GTCAATGAATCGCGT/ACGCGATTCATTGAC | 65 | 28 |
| Module arm (M/M*) | CGCGTTGCGCCTGCC/GGCAGGCGCAACGCG | 79 | 40 |
| Module arm (N/N*) | GAGCCGCGCACGCCG/CGGCGTGCGCGGCTC | 77 | 38 |
| Sacrificial duplex ($D_1$/$D_1$*) | CCGGGCGCGCGGCCG/CGGCCGCGCGCCCGG | 84 | 46 |
| Sacrificial duplex ($D_2$/$D_2$*) | CGTCCGAC/GTCGGACG | 50 | 16 |
| Sacrificial duplex (HP) | CGTCCGACTTTTTTGTCGGACG | 78 | 10 |

[a]Calculated melting temperature, $T_m$, of the corresponding duplex. [b]Free energy of folding. All values were calculated using UNAfold with 10 μM of DNA strands at 100 mM of Na$^{2+}$ and 12 mM of Mg$^{2+}$

the fluorophore to the quencher almost completely prevents the emission of light. Upon stretching, the stress transfer from the DNA matrix into the mechanosensing module induces the unzipping of the small red duplex ($D_x$/$D_x$*, $x = 1$, 2 or HP), resulting in the physical separation of the FRET pair and in an increase in fluorescence (Fig. 1c).

**Assembly of the mechanosensing 3D DNA hydrogels.** DNA hydrogels formed solely by DNA building blocks have been reported based on short ssDNA oligomers[41], ligation of plasmid fragments[42], or by direct enzymatic amplification[43]. Yet, short commercial ssDNA oligomers often yield fragile hydrogels, while the ones produced by enzymatic ligation are hard to functionalize as most of the nucleotides are engaged in DNA duplexes. RCA is a convenient way to prepare ssDNA with controlled repetitive sequences and increased mechanical stability due to their high molecular weight[44], but, their high molecular weight can also be an obstacle for handling. For instance, mixing the hydrogel precursors A and B at room temperature (RT) and at high concentration results in inhomogeneous hydrogel lumps, and heating above the melting transition yields viscous solutions that are hard to be shaped. Therefore, we developed a kinetically controlled protocol that builds upon the slow duplex reorganization below $T_m$ to prepare thermoset-like switch-once pregels, which are liquid at RT but concentrated enough to form gels upon heating (Fig. 2a). To this end, we diluted the two precursor strands, to ca. 0.05 wt%, far below the gelation concentration, before mixing them and annealing the mixture for 5 min at 80 °C. During cooling, the diluted conditions promote the formation of dispersed aggregates (i.e., nano-/microgels), in which the crosslinking domains are engaged in intraparticle duplexes. At RT, that is ca. 40 °C below the $T_m \approx 64$ °C of the duplexes forming the aggregates, the duplexes are frozen, and the colloidal solution can be concentrated by spin filtration over Amicon 30 kDa filters (Merck) to form concentrated, yet freely flowing dispersions (up to ca. 1.4 wt%). Since the concentrated dispersions remain liquid, they can be pipetted and injected into molds for sample shaping, for impregnating a paper, or mixed with particles to form composite materials. After this process, heating to 85 °C melts the crosslinking duplexes and allows for re-organization, and, finally, cooling to RT reforms the crosslinking duplexes that now fix the hydrogel in its final homogeneous and continuous 3D shape. The complete process can be followed by rheology via low-amplitude oscillatory strain-rate controlled temperature ramps, whereby a 10-fold increase in the storage modulus, $G'$, is observed after the heating ramp (Fig. 2b).

We also diluted the pregel dispersions before the thermal fixation step to understand the minimum DNA concentration

needed to form a structurally stable hydrogel. Figure 2c displays the photographs depicting the macroscopic appearance of the DNA hydrogels formed at 0.1–1.4 wt% after the addition of TE buffer. Evidently, all hydrogels with a concentration higher than 0.3 wt% remain physically stable, despite swelling, while lower DNA concentrations result in hydrogels with poor structural integrity.

The rheological characterization of the resulting hydrogels shows the evolution of the mechanical properties of the hydrogels with temperature (Fig. 2d). As observed macroscopically, an increase in the DNA concentration leads to a higher mechanical strength of the hydrogels. At RT, the storage modulus, $G'$, of the hydrogel spans from as low as 0.4 Pa for 0.35 wt% up to above 400 Pa for a 1.4 wt% DNA. This is a direct result of providing a higher crosslinking density in a given volume during the re-organization step in the temperature ramp. The values are similar to the ones reported for plasmid-based hydrogels[42]. The mechanical properties remain relatively stable between RT and about 60 °C, where the $T_m$ (ca. 64 °C) of the duplex crosslinks is located. At these temperatures, the duplex crosslinks become dynamic (at $T = T_m$, 50% of the duplexes are molten), which allows for network reorganization and energy dissipation during shear deformation. The strong increase of the loss modulus, $G''$, close to the $T_m$, is characteristic of a dynamic network with transient supramolecular bonds. Heating further above 70 °C results in the complete melting of the DNA hydrogel and in a strong decrease in loss and storage moduli, corresponding to the onset of the gel/sol transition. Above 75 °C, the crosslinking duplexes are almost fully molten, and the material behaves as a liquid. The change in $G''$ is less pronounced for the highest DNA concentration (1.4 wt%), because the high concentration of ssDNA polymer leads to a strongly entangled, highly viscous polymer solution. Except for the first thermosetting heating cycle, the temperature-dependent rheological behavior is highly reversible, indicating a thermo-reversible all-DNA hydrogel material that can allow for shaping and recycling on demand (Fig. 2e).

Functionalization of the DNA hydrogel with stoichiometric amounts of a force-sensing module, taking $D_1$ as an example, goes along with a swelling of the hydrogel, which counterbalances the introduction of further crosslinks and consequently does not alter the mechanical properties of the hydrogel at a low amplitude (Supplementary Note 2, Supplementary Fig. 2b). Yet, the effects of network modification are visible by the decrease of storage modulus at lower amplitudes for functionalized hydrogels than for pristine ones (Supplementary Fig. 2a), and, in close relation, the melting transitions during the temperature ramps get more complex after functionalization (Supplementary Fig. 2c). Note that we always performed the functionalization of the hydrogels at a low temperature using already folded mechanosensing

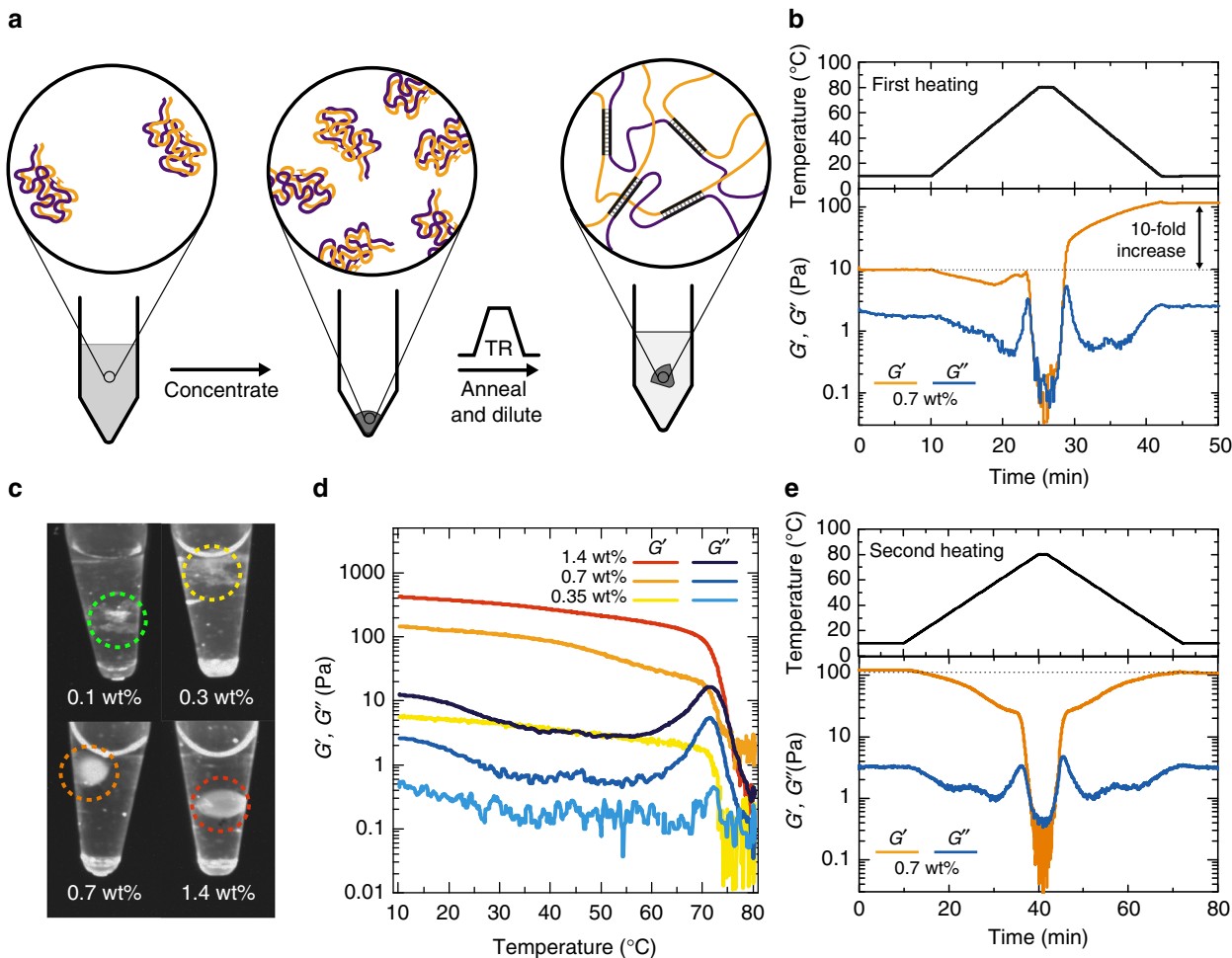

**Fig. 2** Thermosetting and thermo-reversible DNA hydrogels. **a** Schematic representation of the two-step DNA hydrogel assembly. The last dilution step allows for a check of structural integrity. **b** Rheological properties of a 0.7 wt% dispersion upon heating to 80 °C and cooling to RT. Heating above the $T_m$ of X/X* enables the re-organization of the supramolecular duplex and the formation of a continuous hydrogel network and yields a 10-fold increase in storage modulus. **c** Macroscopic black and white fluorescent images of hydrogels prepared with 0.1 wt% up to 1.4 wt% of DNA (functionalized with $Atto_{488}$-labeled ssDNA oligomers for visualization) and re-suspended in TE buffer (NaCl 100 mM, $MgAc_2$ 12 mM; the hydrogels are encircled with dashed lines). The hydrogels are stable above 0.7 wt% and disintegrate below 0.3 wt%. **d** Temperature-dependent rheological properties (frequency 1 Hz, strain amplitude 5%) of DNA hydrogels at different concentrations. **e** Rheological properties of an already formed 0.7 wt% hydrogel upon heating to 80 °C and cooling back to RT. After the first heating cycle, the hydrogel behaves as a thermoplastic, which melts above the $T_m$ of X/X*(ca. 64 °C). RT, room temperature

modules. This process reduces the possibilities of module misfolding and unwanted reorganization, thereby minimizing the residual fluorescence at rest. Hence, such a functionalization strategy with mechanosensing modules has minimal impact on the material mechanical properties.

**Characterization of the mechanosensing 3D DNA hydrogels**. For the quantitative characterization of mechanosensing, we focused on hydrogels formed at 0.7 wt%, which minimizes the amount of DNA needed combined with easy processing and good mechanical stability. The mechanosensing specimens are simply prepared by pipetting the gel premix (ca. 1.3 µL) in custom-designed specimen molds. The molds are cut from parafilm and form a 1 × 5 mm chamber with a 1-mm paper overhang at each end to fix the hydrogel (Supplementary Fig. 3). The gel premix in the molds is sandwiched between a glass slide and a polyurethane film and heated for 2 min at 85 °C on a Peltier-controlled heating chamber. After cooling down to 4 °C for 10 min, the supports are separated and the gel suspended on its mold is placed in a PCR tube containing 1 mL TE buffer (NaCl 100 mM, $MgAc_2$ 12 mM).

A typical hydrogel contains about 200 pmol of each repeat unit of the two ssDNA hydrogel precursors. Finally, stoichiometric amounts of pre-assembled mechanosensing modules and reference fluorophore (100 pmol each) are added to the tube, corresponding to ca. 50% of the available barcode domains, and everything is left shaking gently overnight at 15 °C to hybridize (see Methods, Supplementary Figs. 3, 4 for details on the mechanosensing module assembly and sample preparation). An example of a resulting film in its custom-designed sample holder is presented at the top of Fig. 3b. The used force-sensing modules are summarized in Fig. 3a and Tables 1, 2 and vary by the nature of the sacrificial bond in pink. $D_1$ and $D_2$ present sacrificial duplexes with different lengths, HP has the same duplex sequence as $D_2$ but the two ends are connected by a $T_6$ loop to form a HP loop, while T and N are controls with a covalent $T_6$ link (without sacrificial bond) and without any connections, respectively.

Figure 3b presents wide-field fluorescence microscopy during the stretching of a hydrogel functionalized with the module $D_1$ (strain rate ca. 3 mm/min). As expected, the film is strongly mechanofluorescent and a convincing increase in red fluorescence ($\lambda_{ex} = 572$ nm; $\lambda_{em} = 629$ nm) appears during stretching,

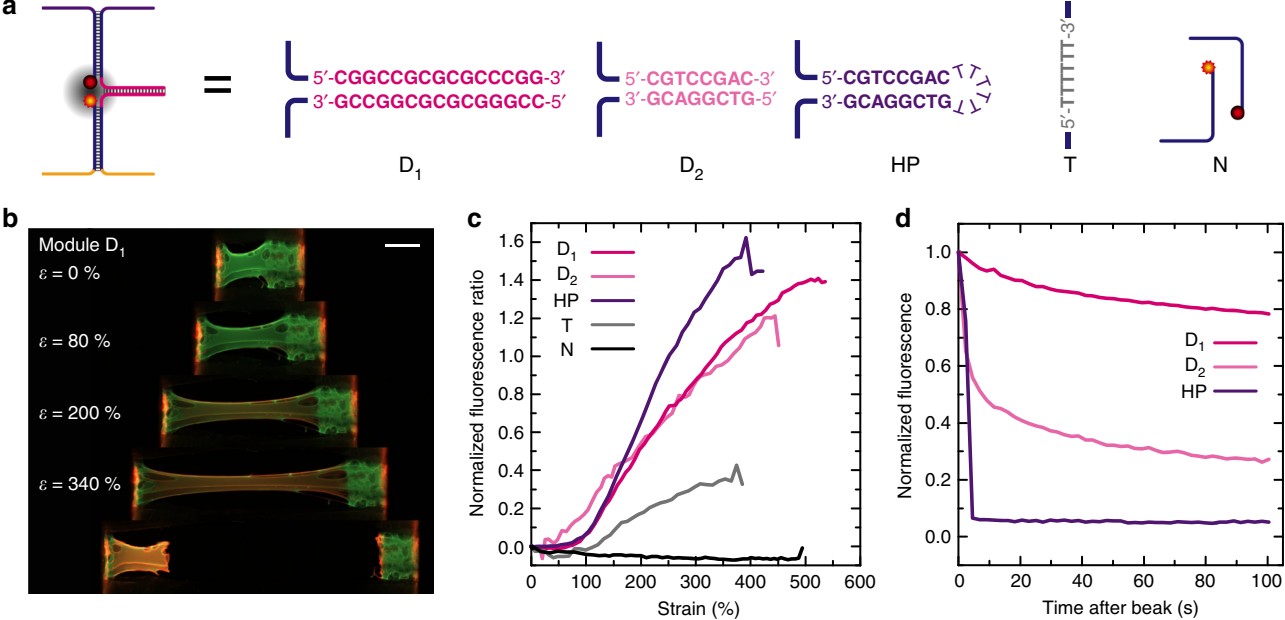

**Fig. 3** Strain and temporal response of the mechanofluorescent DNA hydrogels with different force-sensing modules. **a** Sequence and structure of the different modules investigated. Note that the rest of the modules, fluorophores (Atto$_{488}$, Atto$_{565}$), quencher (Iowa Black RQ), and attachment strands are identical in all experiments. The T and N modules are controls in which the fluorophore and quencher strands are (T) connected via a $T_6$ covalent link without sacrificial duplex or not connected (N) at all. **b** Fluorescence imaging at different elongations of a 0.7 wt% hydrogel functionalized with $D_1$. Scale bar = 1 mm. **c** Evolution of the red/green fluorescence ratio $R/G(\varepsilon)$ after subtraction of $R/G_{ini}$ for the different DNA hydrogels functionalized with the mechanofluorescent modules. **d** Temporal recovery of the fluorescence ratio after hydrogel failure (normalized to $R/G_{max}$ and $R/G_{ini}$)

while the green reference fluorescence ($\lambda_{ex} = 485$ nm; $\lambda_{em} = 540$ nm) decreases due to stretching-induced thinning of the hydrogel. The increase of the red fluorescence stems from the mechanosensing module, which is activated upon stretching the DNA hydrogel and by transferring the stress into it. This leads to an unzipping of the $D_1$ duplex and concurrent separation of the Atto$_{565}$-functionalized strand from the quencher-functionalized one (Fig. 1c), resulting in a FRET decrease and increase of fluorescence. The decrease of fluorescence observed for the green channel is purely a geometric effect, because the areal density of the Atto$_{488}$ fluorophore, which is bound to the DNA network, decreases upon stretching. This also illustrates the need of a reference fluorophore that is not mechanosensitive to allow for a more in-depth analysis. By plotting the ratio of red and green fluorescence, it is possible to balance these geometrical effects of stretching and extract a quantitative measurement of the fluorescence increase (Fig. 3c).

We build upon the modular approach of DNA design to explore the different architectures of the force-sensing module, incorporating sacrificial bonds and various controls (Fig. 3d). We designed two duplexes with a similar composition but with different lengths and melting transitions ($T_{m, D1/D1^\star} = 84$ °C and $T_{m, D2/D2^\star} = 50$ °C), as well as a HP duplex with the same sequence as $D_1$, but with a covalent connector loop at the end ($T_{m, HP} = 78$ °C). We also present two control modules, one in which the sacrificial duplex is replaced by a short ssDNA strand that cannot open (T) and another one without any bond to bridge the FRET pair (N). Temperature-dependent fluorescence measurements of the individual modules confirm the distinctly lower transition for module $D_2$ (Supplementary Note 5, Supplementary Fig. 5b). As expected, the modules $D_1$, $D_2$, and HP, which present a sacrificial duplex, show a strong mechanofluorescent response with a steady increase of red fluorescence upon stretching (Fig. 3c). The control module N already shows a strong red fluorescence at rest, which does not increase upon stretching.

Hence, a flat line is obtained upon normalization to the green background fluorophore, as the two dyes change in observed emission similarly during stretching. This negative control confirms that bridging of the FRET pair is necessary to observe a fluorescent response from stress sensing. The somewhat surprising increase of fluorescence for module T, the directly connected motif without sacrificial duplex nor hairpin, originates from the opening of the module arm M or N due to strain-induced unwinding—a phenomenon known from single-molecule experiments[45]. Note that partial unwinding of DNA duplexes in in-line tensions requires higher forces (around 70 pN) than unzipping DNA duplex from an end with perpendicular tension (5–20 pN), hence explaining the much weaker response observed for module T[45,46]. This additional control critically confirms that the strong increase in fluorescence observed for modules $D_1$, $D_2$, and HP results indeed from duplex unzipping from the weak opening point.

The modules presenting a synthetically encoded mechano-fluorescent response ($D_1$, $D_2$, HP) yield similar curves with an absence of fluorescence response below 100% strain and a steep increase above (Fig. 3c). Note that we performed at least three measurements for each module (Supplementary Note 6, Supplementary Fig. 6). $D_2$ shows an earlier onset at 30% elongation, rather than 100% for the other motifs, which is expected due to its lower free energy and lower melting transition (Table 2 and Supplementary Fig. 5). Since module $D_2$ requires less force and work to break than X/X$^\star$, it breaks first and shows fluorescence earlier than any other module. On the contrary, both $D_1$ and HP have higher $T_m$ than $D_2$. The similar threshold in the fluorescence increase originates from the fact that low-energy conformational rearrangements (uncoiling of single-stranded domains and rupture of X/X$^\star$) enable strain accommodation first in the ssDNA polymers without module rupture, as also observed for other mechanochromic materials[23]. At a low strain, these rearrangements are sufficient to accommodate the hydrogel

stretching, while the forces transferred to the mechanosensing modules are yet insufficient to induce a response. Above the critical stretching ratio, the network rearrangements are insufficient to compensate the stretching, and the force-sensing modules located along the shortest linkages between the crosslinking points start to open. Since $D_1$ and HP (and T) are stronger than X/X* bonds, the onset of fluorescence takes place at the same strain when the support network cannot reorganize further. Further stretching induces a supramolecular rearrangement and stress transfer from one short link to the next one and results in a steady fluorescence increase as more and more of the mechanofluorescent sensory modules open. Since we work with non-perfect networks, it is the presence of various short linkages between the mechanosensing modules that governs the behavior. Hence, during stretching, the proposed design allows primarily to engineer the onset of fluorescence increase, as weaker modules require less energy to open, leading to materials with lower critical strains.

On the contrary, the architecture of the module, i.e., whether composed of two strands ($D_1$, $D_2$) or covalently linked via an HP or linear linker (T), plays a critical role in the temporal response of the mechanofluorescent hydrogel during stress relaxation. Figure 3d presents the fluorescence recovery after gel failure. The hydrogel functionalized with the HP module comes back immediately to the initial state (before stretching), while hydrogels functionalized with $D_1$ or $D_2$ return slowly, but, in fact, can never fully recover their original configuration. The short covalent $T_6$ link in the HP module prevents complete destruction and global rearrangement, and maintains the two parts of the FRET pair (Atto$_{564}$ and quencher) at close proximity. After stress release, the HP immediately reforms and the fluorescence drops as the module returns to its original state. In modules $D_1$ and $D_2$ on the other hand, there is no covalent linkage between the two parts of the module and once the two single strands release each other, a full rehybridization requires diffusion and more complex reorganization, which in turn requires longer time scales. The slower recovery of the $D_1$ duplex compared to $D_2$ is less intuitive. It can, however, be explained by the fact that after opening $D_1$, the constituting strands can form a metastable duplex with their own kind (Supplementary Note 7, Supplementary Fig. 7). This further impedes the re-formation of the original module and fluorescence decrease. DNA-based mechanofluorescent hydrogels therefore offer an easy access to control the temporal stress-relaxation behavior, from fully reversible to transient or near irreversible.

**Application of the mechanofluorescent 3D DNA hydrogels.** The use of a thermo-reversible DNA hydrogel matrix allows for an easy modulation of the matrix architecture to increase, e.g., the local strain that the mechanochromic module feels, which increases the overall strain sensitivity of the material similarly as biological force-sensing modules[9]. We showcase this effect with cellulose fiber-DNA composites prepared by impregnating a piece of paper with the DNA pregel, applying the standard heating ramp, and functionalization of the DNA gel with the $D_1$ module. The high-modulus cellulose fibers form a rigid network with low strain at break, which interpenetrates the soft and viscoelastic force-sensing DNA matrix. As visible in Fig. 4a, stretching results in a strong fluorescence increase at already ca. 10% global strain. The early rupture of the cellulose fiber network that bears most of the stress results in a high stretching ratio for the DNA matrix near the rupture line, which induces a strong fluorescence increase at low strain. This cellulose/DNA composite system, which could be further expanded to different materials, serves as an instructive example of how to impart DNA-based

mechanofluorescence to composite materials for monitoring their structural integrity.

We further show that our mechanofluorescent DNA hydrogels can reveal insights into subtle mechanical behaviors and complex strain fields by using confocal laser scanning microscopy (CLSM) for multiscale 3D analysis. Figure 4b presents a microcomposite prepared by dispersing 5 μm sulfate-functionalized polystyrene (PS) microspheres in a DNA hydrogel matrix functionalized with the $D_1$ module. CLSM, immediately after stretching, visualizes areas of high stress at different length scales. At the mesoscale, a strong red fluorescence appears along a crack line on the right-hand side, while it is possible to distinguish an inhomogeneous red fluorescence around the PS microspheres with stronger fluorescence along the stretch direction. Here the strong red fluorescence along the particle sides, which are parallel to the stretching direction, indicates the absence of interfacial binding between the particle surface and the matrix (as depicted on the scheme). This is indeed expected in the case of PS beads with a negative surface charge due to charge repulsion with the negatively charged DNA. Yet it demonstrates that mechanochromic DNA matrices open access to key microstructural failure mechanisms, which are hardly accessible by other means, and which could even be followed in 3D using CLSM.

Finally, we show that our mechanosensitive DNA hydrogels are able to reveal freezing-induced localized stress patterns, which would be hard to access by other means. To this end, a strip of the DNA hydrogel functionalized with the $D_1$ module was placed in TE buffer solution and quickly frozen in liquid nitrogen (within ~1 min), which leads to ice-templated growth of the structures[47]. After complete freezing of the entire buffer solution, the tube was left at RT to unfreeze, and the resulting hydrogel was imaged using wide-field fluorescence microscopy (Fig. 4c). The mechanosensing DNA hydrogel shows regularly spaced fluorescence patches, which reveals that the growth of ice crystals in the hydrogel induces high-stress patterns localized regularly along the ice-growth direction. Undoubtedly, such a mechanosensing matrix can be further applied to other systems of interest to understand strain distribution in complex and difficult-to-probe structures.

## Discussion

We have introduced the facile assembly of thermosetting and thermo-reversible DNA hydrogels of high stretchability and demonstrated the integration of a modular toolbox of DNA tension probes to realize programmable mechanofluorescent properties. Functionalization with various programmable force-sensing modules and quantitative strain monitoring in the material reveal that the mechanically triggered fluorescence increase is primarily controlled by the topology of the hydrogel network and less by the mechanophore architecture. This motivates us to pursue perfectly structured hydrogel networks in future to synchronize the activation of all mechanophores and decipher detailed differences in their behavior. In stark contrast, the mechanophore response during stress relaxation is strongly related to its geometry. Hairpin structures, in which both ends of the sacrificial sensor duplex are linked together, quickly close back and recover their unstretched configuration upon stress release. Sacrificial duplexes without a covalent link at their center can end up much farther apart during force-induced rupture, which slows down the recovery.

We showcased the applicability and potential of our fluorescent mechanochromic matrix as a tool to investigate complex strain fields in homogeneous and composite materials. We demonstrated multiscale 3D mapping of strain fields in composite hydrogels and revealed the presence of localized freezing-induced strain patterns

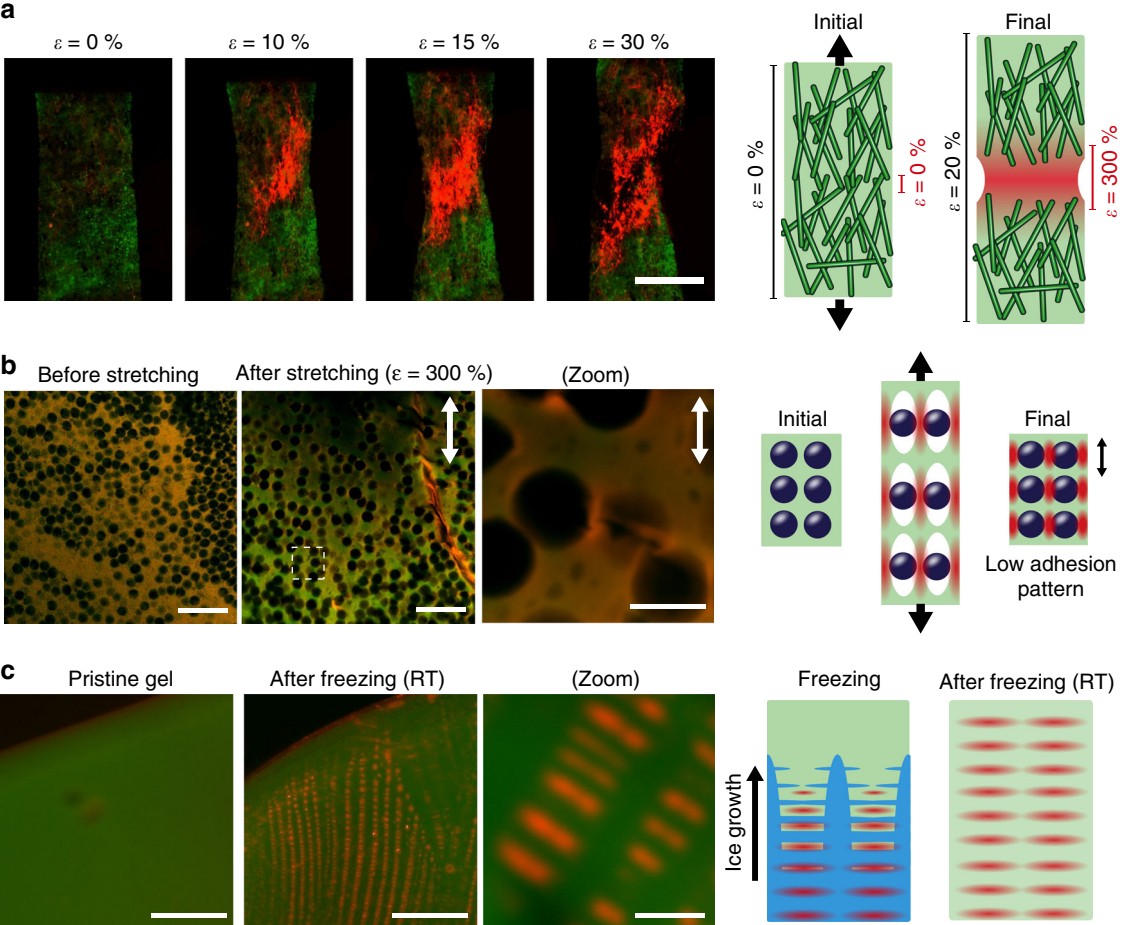

**Fig. 4** Fluorescent mechanosensing enables visualization of complex strain-field inhomogeneities. **a** Composite materials with interpenetrated cellulose microfiber (paper) and mechanosensitive DNA hydrogel network incorporating the $D_1$ module show increased strain sensitivity. The stress focusing from the rigid cellulose fiber network into the flexible DNA hydrogel along the rupture line of the cellulose fiber network results in a strong fluorescence increase at a low global strain. Scale bar = 1 mm. **b** Combination of CLSM imaging and mechanofluorescent DNA matrix enables multiscale visualization in a PS microsphere-loaded mechanosensitive DNA hydrogel network incorporating the $D_1$ module. Here the microscale stress pattern reveals a poor adhesion of the DNA matrix onto the negatively charged microsphere surface. Note that the contrast has been adjusted to enable visualization of the absence (before stretching) or presence (after stretching) of local variation of fluorescence. Scale bars = 25 μm (left, center) and 5 μm (right). **c** The mechanosensitive DNA hydrogel network incorporating the $D_1$ module reveals regular strain patterns caused by ice-crystal formation during freezing. Scale bars = 250 μm (left, center) and 50 μm (right). CLSM, confocal laser scanning microscopy; PS, polystyrene

in hydrogels. The modularity of our system to program reversible and irreversible mechanofluorescent behavior as well as the possibility to quantify strain in complex materials open diverse possibilities to study complex materials at previously inaccessible programmability of force and response behavior in 3D.

## Methods

**Rolling circle amplification**. The overview of the process is presented in Supplementary Fig. 1 together with the gel electrophoresis of the starting (linear RCA templates) and final materials (RCA product).

First, the template and its corresponding ligation strand were mixed to reach a final concentration of 5 μM in TE buffer containing an additional 100 mM NaCl (total 100 μL). The solution was heated up to 85 °C and cooled down to 25 °C at 0.01 °C s$^{-1}$. After hybridization, 20 μL of 10× commercial ligase buffer (Lucigen: 500 mM TRIS-HCI, 100 mM MgCl$_2$, 50 mM dithiothreitol, and 10 mM ATP), 70 μL of water, and 10 μL of T$_4$ ligase (2 U μL$^{-1}$) were introduced into a tube containing 100 μL of the template strand, gently mixed, and left to react for 3 h at RT. The enzyme was then deactivated by heating the mixture for 20 min at 70 °C.

Then, 10 μL of exonuclease I (40 U μL$^{-1}$) and 10 μL of exonuclease III (200 U μL$^{-1}$) were introduced and the mixture was left overnight at 37 °C to remove ligation strands and non-circularized templates in solution. The enzymes were then deactivated by heating at 80 °C for 40 min. The templates were purified by filtration over Amicon Ultracentrifugal filters with 10 kDa cut-off (Merck Millipore) and rinsed three times using TE buffer in the same filter. The ssDNA concentrations were measured using a

ScanDrop (Analytik Jena) spectrophotometer, and the solutions were diluted to 1 μM using TE buffer.

For RCA, 50 μL of circularized template (1 μM) was mixed with 760 μL of ultrapure water, 10 μL of exonuclease-resistant primer (10 μM in TE buffer), 100 μL of commercial 10 × polymerase buffer (Lucigen: 500 mM TRIS-HCl, 100 mM (NH$_4$)$_2$SO$_4$, 40 mM dithiothreitol, 100 mM MgCl$_2$), 20 μL of Φ$_{29}$ Polymerase (10 U μL$^{-1}$), and 1 μL of pyrophosphatase (2 U μL$^{-1}$). The mixture was left for 1 h at RT in order to pre-form the enzyme DNA complexes before the addition of 50 μL of an adjusted dNTP mix (total dNTP concentration of 100 mM; the proportion of each base corresponds to the expected product composition) to start the amplification reaction. The samples were kept for 60 h at 30 °C before inactivation for 10 min at 80 °C. The product was then concentrated by filtration over Amicon Ultracentrifugal filters with 30 kDa cut-off (Merck Millipore) and rinsed three times using 400 μL of TE buffer. The amount and purity of the ssDNA concentration were then measured using a ScanDrop (Jena Analytic) spectrophotometer using 50 times diluted solutions and 33 μg per OD$_{260}$ as the standard absorbance coefficient for ssDNA.

It is important to add pyrophosphatase to the mixture to avoid the formation of DNA/MgP$_2$O$_7$ nanoflowers. Note that while nanoflowers were initially thought to consist of pure ssDNA, but they are actually formed through the complexation of ssDNA with magnesium pyrophosphate crystals, a by-product of the nucleoside triphosphate polymerization[40,43,48,49].

**Pregel solution preparation**. The DNA solutions of the RCA products A and B were diluted separately to about 0.5 g L$^{-1}$ (1 mL) in TE buffer containing 12 mM of MgAc$_2$ and 100 mM of NaCl, and the resulting mixture was heated to 85 °C for

5 min for homogenization. The concentration of each hydrogel precursor was measured by UV–vis spectroscopy and both strands were mixed together in stoichiometric proportion at RT and left to hybridize for 1 h at 4 °C (Supplementary Fig. 1b, lane 5). About 2 mL of the resulting mixture was concentrated by filtration over Amicon Ultracentrifugal filters with 30 kDa cut-off (Merck Millipore) down to 60 µL of solution (10 min at 15,000 g) and 6 µL of 1 wt% Tween 80 solution in water was added to facilitate manipulation. The resulting solution was a fluid and contained about 1.6 wt% of DNA as measured by UV–vis spectroscopy. The pregel solution was diluted to 0.7 wt% using TE buffer containing 100 mM of NaCl, 12 mM of MgAc$_2$, and 0.1 wt% of Tween 80 for most gel preparations.

**Assembly of mechanofluorescent sensor modules.** The different polyacrylamide gel electrophoresis-purified oligomers forming the mechanosensing modules (e.g., for module D$_1$: Atto$_{565}$-MA$_4^*$, B$_4^*$N-IowaBQ, A$_1^*$M$^*$D$_1^*$, and D$_1$N$^*$B$_1^*$ dissolved in TE buffer at 100 µM) were mixed with 10 × folding buffer (containing 120 mM MgAc$_2$ and 1 M NaCl) and completed with TE buffer to reach a final concentration of 20 µM of module (i.e., 20 µM in TE buffer containing 100 mM of NaCl and 12 mM of MgAc$_2$). In order to minimize the fluorescence at rest, the quencher strand was introduced in slight excess (1.1eq.) and the fluorophore strand in slight lack (0.9eq.) compared to the sacrificial duplex strand. This ensures that each fluorophore ends up in a module containing a quencher. The mixture was heated to 85 °C and cooled down to 25 °C at 0.01 °Cs$^{-1}$ to form the module.

We checked the correct assembly of the modules by gel electrophoresis (Supplementary Fig. 4). The module components were mixed in TE buffer containing 100 mM of NaCl and 12 mM of MgAc$_2$, heated to 85 °C, and cooled down to 25 °C at 0.01 °Cs$^{-1}$ before loading in each lane. In Supplementary Fig. 4a, the modules are folded using the quencher (IowaRQ)-functionalized strands, which impedes the observation of a fully folded module due to the quenching of DNA stains (lanes 4 and 7). It is possible to overcome this problem by using a non-functionalized B$_1^*$N strand, as visible in Supplementary Fig. 4b (lane 5). Incomplete modules tend to form various self-dimers that appear as bands with a higher size (typically a multiple of the single component size). These bands are absent for the complete modules, as the sequences leading to these self-dimers are now engaged in duplexes with a higher binding affinity. Another way of checking the structural integrity of the folded module is to look at the remaining fluorescence after self-assembly, which is very low when the modules are properly folded (see also Supplementary Fig. 4).

**Sample-holder preparation.** The sample holders were prepared by regularly cutting 14 individual 1 × 4 mm slits into two 75 × 26 mm Parafilm M (Bemis Inc.) strips. Then, two 5 × 70 mm strips of Precision Whipe dust-free paper (Kimtech Science, Kimberly Clark$^{TM}$) were sandwiched between two of the pre-cut Parafilm strips as described in Supplementary Fig. 3a, b. The slits of the two Parafilm strips need to superimpose exactly, and the two Precision Whipe paper strips were carefully placed so that about 1 mm of the paper looked out at each end of the slits. The construct was placed between two slightly greased (Glisseal HV grease; Borer Chemie) microscope glass slides (75 × 26 mm) and heated to 90 °C on a hot plate to melt the parafilm together with the paper strips and form the final array of holders. After cooling for 5 min, the glass slides were separated and the sample holders were released (Supplementary Fig. 3c, d). Finally, the 14 individual holders were manually cut using scissors (Supplementary Figure 3e). The holders were examined and the mis-shaped ones or holders without accessible paper films were discarded.

**Free-standing DNA hydrogels for strain tests.** To prepare the free-standing DNA hydrogels, about 1.3 µL of liquid pre-gel solution at 0.7 wt% (containing 155 µM of both A$_1$A$_2$XA$_3$A$_4$ and B$_1$B$_2$X$^*$B$_3$B$_4$ repeat template strands) was placed in the slit of a sample holder resting on a glass slide, and a soft polyurethane cap (Clear Flex 30, 10 × 5 mm and ~ 1 mm thick) was carefully placed on top of the slit (air bubbles were avoided as much as possible; Supplementary Fig. 3f). The filled holder was placed in a thermocycler and heated to 80 °C for 2 min before leaving it at 4 °C for 10 min. The thermal treatment induces gel formation, which remains entrapped in the paper at each end of the slit. After removing the polyurethane cover, the sample holder was peeled off from the glass slide and placed inside a 0.5 mL Eppendorf tube containing 300 µL of TE containing 12 mM of MgAc$_2$ and 100 mM of NaCl.

**Functionalization of the DNA hydrogels with force sensors.** For functionalization, 5 µL of mechanofluorescent sensory modules (20 µM) and 1 µL of Atto$_{488}$-A$_3$ (100 µM) were introduced in the tube containing the piece of suspended hydrogel immersed in TE buffer containing 12 mM of MgAc$_2$ and 100 mM of NaCl. The samples were left softly shaking overnight at 4 °C before performing the stretching experiments.

**Composite DNA hydrogels.** Similarly, the cellulose paper/DNA hybrid structures were formed by soaking 1 × 5-mm wide precision Whip dust-free paper (Kimberly Clark$^{TM}$; Kimtech Science) and submitting it to the same thermal treatment under a polyurethane cover to prevent water evaporation. For the composite, 5 µm sulfate-functionalized polystyrene microspheres (10 wt% in water) were mixed in 1: 1 ratio with the liquid hydrogel premix at 1.6 wt% and placed directly in the sample

holder, followed by a temperature ramp for setting the hydrogel prior to mechanical tests.

**Tensile tests with in situ fluorescence monitoring.** Imaging was performed using a AxioZoom widefield microscope with a 1 × objective lens (Aperture 0.25) sequentially imaging the green channel (Filter set 38 HE: excitation 440/470, emission 525/550; exposure 200 ms) and the red channel (Filter set 63 HE: excitation 572/525, emission 629/662; 500 ms exposure) resulting in 0.44 image/s with filter changes. Stretching was performed using a screw-driven custom-built extensometer allowing the DNA hydrogel to be immersed in TE buffer supplemented with 100 mM NaCl and 12 mM MgAc$_2$ during the experiment. The buffer solution was regularly changed with ice-cooled fresh buffer to maintain the sample below 20 °C. Stretching was performed at a rate of ca. 3 mm min$^{-1}$.

**Temporal recovery experiments.** For the temporal recovery experiments, the DNA hydrogels were continuously imaged for about 100 s after gel rupture. The ratios of red to green fluorescence were measured on the same small area as for the stretching experiment and were normalized to the minimal (before stretching) and maximal (immediately before rupture) fluorescence ratio.

**Image treatment and analysis.** The series of fluorescence images were treated using ImageJ by first performing a background cleaning (200 pixel radius). Then, the temporal profiles of both the red and green channels in the region of interest were extracted. The values plotted correspond to the red/green ratio after subtraction of the initial R/G ratio. The strains were calculated based on the sample lengths measured manually every five frames of the videos.

We systematically measured the fluorescence ratios on a small area at the center of the tensile specimen (in yellow), which focuses on a section of the sample that is actually stretched and grants maximal sensitivity (Supplementary Fig. 6a–c). We also measured the fluorescence over the full area (in purple), which confirms the overall increase of red fluorescence and the absence of bleaching as the level of green fluorescence remains constant during the entire stretching experiment (Supplementary Fig. 6b). This method allows obtaining reproducible and quantitative strain/fluorescence curves for all systems as presented in Supplementary Fig. 6d–g for the different modules, as well as for the temporal recovery experiments (Supplementary Fig. 6h).

## Data availability
The data that support the plots within this paper and other findings of this study are available from the corresponding author upon reasonable request.

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

## Acknowledgements

This work was funded by the ERC Starting Grant "TimeProSAMat" (677960). G.D. and L.H.C acknowledge financial support from FAPESP grant number 2016/22778-1 and 2011/21442-6. We thank Simon Ludwanowski for the help with the spectroscopy setup. A.W. appreciates discussions within the DFG Cluster of Excellence livMatS team.

## Author contributions

R.M. and A.W. conceived the project and designed the experiments. R.M., G.D. and L.H. carried out the experiments. R.M. and A.W. analyzed the data. A.W. supervised the project. All authors discussed the results and commented on the manuscript. R.M., L.C. and A.W. wrote the manuscript.

## Additional information

**Competing interests:** The authors declare no competing interests.

