## [Peer Review File · Nature Communications]

Reviewers' comments:

Reviewer #1 (Remarks to the Author):

The authors presented mechanosensing all DNA hydrogels by implementing of FRET based detection using DNA force probes. Although FRET based force sensing and detection of DNA dehybridization is not new, this work represents an interesting trial on the modular design of all DNA hydrogels that can offer spatiotemporal resolution as well as programmable sensitivity and reversibility of mechanosensing. This work may be suitable for publication in Nature communications. However, I suggest the following considerations:

1. The authors put a lot efforts to the description of the DNA hydrogels without functionalization with the force-sensing and control modules. However, the more important part is the hydrogels with the force-sensing and control modules. It is more important to have the rheological or tensile measurements on the functionalized hydrogels. Also, I feel it is highly valuable to study the thermal stability of the different force-sensing modules and the X/X* crosslinking domain and even the barcode domains (e.g., thermally induced gel-sol transition and FRET of the functionalized hydrogels), because this is important to the interpretation of force responses of different moieties (relative strength).
2. The interpretation of the threshold in the fluorescence increase needs some reconsideration. For example, for the hydrogel with D1, D1 is stronger than X/X* crosslinking domain. Therefore, it is highly possible that under tensile stress, X/X* crosslinking domain will dissociate earlier than D1, leading to larger strain threshold. And due to the dissociation of X/X* crosslinking domain, it may take longer time for D1 to reform.

Reviewer #2 (Remarks to the Author):

The manuscript describes modular design of programmable mechanofluorescent DNA hydrogels. Sensitivity and modularity of mechanosensitive materials are known to be difficult to tune and/or to achieve. The authors could successfully design DNA modules to programmable hydrogels. The design has been precisely described, and the results are elegant. Several experimental supports are enough to admire the achievements. Figures 3 and 4 strongly support the concept. There are no points to be revised, and thus, the reviewer recommends the manuscript to accept without further revisions.

Reviewer #3 (Remarks to the Author):

The manuscript entitled "Modular Design of Programmable Mechanofluorescent DNA Hydrogels" should be rejected based on the following reasons.

1. There is really nothing new in terms of technical logical innovation. Every single aspect of the technical designs is not new.
2. The significance of the research outcome, in this case, the development of a fluorescent DNA Hydrogel sensitive to stretch, is unclear. The couple of examples shown in this study are unconvincing. In other words, there is no practical utility of this gel.
3. A major problem of this gel is that the color change is not correlated to physical distance displacement from the original position. As a result, talking about "programmable" is unfounded. There is nothing programmable at all. The outcome is not quantitative.
4. The sensitivity of the mechanofluorescent DNA hydrogels is at best low. As shown in Figure 3, the distance change due to stretching must be huge before a color change can be seen or clearly detected. By then, however, the gel is broken. There is never a quantitative relationship between the color change and distance displacement. Hence, what is the use of this gel?
5. The rheological study of these gels is grossly inadequate. For instance, the frequency sweep

experiment should have been done, and temp should be changed as well. GN should have been done as well.

6. Finally this reviewer has a lot of issues with the improper use of terminology and misleading terms or descriptions. Below are just some glaring examples.

(1). The title is misleading. The fluorophore is not mechanosensitive at all. Further, the fluorescent intensity is not programmable. The authors have not done enough to show this can be programmable or "tunable" (as in the last paragraph on page 3).

(2). The authors do not do FRET measurements at all; yet they stated so in the Abstract. A color change is not a FRET-based detection at all.

(3) The use of "mechano" as the prefix to describe the gel is misleading. The gels containing the "force sensor modules" can only respond to tension. The authors have not shown any other type of forces that can cause mechanical movement or displacement.

(4) Fig 1 is fine to illustrate the idea. But there are lots of problems in this figure. (a) The use of "crosslinks" is troubling. Where is the cross-linker? (b) The nature of the bond that governs DNA base-pairing is hydrogen bond, not sacrificial bonds or duplex. It is just wrong. (c) The authors cannot rule out a heterogeneous formulation of the so-called "bar codes" between precursors A and B. Perhaps C and D may be involved as well. This may be true especially when precursor concentrations becomes higher. The so-called mechanofluorescent module concentration could further contribute to the complexity of the polymer network structures. Yet, the authors have neither done a detailed study nor acknowledged these problems. These sources of heterogeneity could seriously affect the mechanical stability of the gels.

Reviewer #1 (Remarks to the Author):

The authors presented mechanosensing all DNA hydrogels by implementing of FRET based detection using DNA force probes. Although FRET based force sensing and detection of DNA dehybridization is not new, this work represents an interesting trial on the modular design of all DNA hydrogels that can offer spatiotemporal resolution as well as programmable sensitivity and reversibility of mechanosensing. This work may be suitable for publication in Nature communications. However, I suggest the following considerations:

We thank the reviewer for his/her remarks which helped to improve the manuscript. We now added supplementary characterization of the melting behaviour of the modules, as well as rheological characterization of the hydrogels after modification. We also provide, in the main text, additional references to FRET-based force sensing and we revised some of the discussion on the mechanical behaviour of the hydrogels.

1. The authors put a lot efforts to the description of the DNA hydrogels without functionalization with the force-sensing and control modules. However, the more important part is the hydrogels with the force-sensing and control modules. It is more important to have the rheological or tensile measurements on the functionalized hydrogels. Also, I feel it is highly valuable to study the thermal stability of the different force-sensing modules and the X/X* crosslinking domain and even the barcode domains (e.g., thermally induced gel-sol transition and FRET of the functionalized hydrogels), because this is important to the interpretation of force responses of different moieties (relative strength).

We now provide a rheological comparison of the DNA hydrogel before and after functionalization with module D1 in the supplementary information (supplementary Figure S5). We observed only minor rheological changes upon functionalization with module D1, which is due to swelling of the hydrogel when functionalizing with D1 (by infusion of a solution), which in turn compensates the increase of crosslinking density.

We also provide comparison of the thermal stability of the different force-sensing modules via temperature-dependent fluorescence spectroscopy (Supplementary Figure S4). The modules D₁, HP and T melt around 75°C while module D2 melts around 45°C. We conclude that the increase of fluorescence observed for D1, HP and T result from the combined melting of the attachment arm (M and N) and of the sacrificial bond which all melt in the same temperature range. These measurements confirm that D2 is indeed weaker and opens at lower temperature than other module. Although the fluorescence measurements fail to distinguish bond strength among D1, HP and T, the similarity of melting and dissimilarity of mechanical response within these modules and their corresponding materials, respectively, underline the importance of geometry to design mechanofluorescent force modules. We implemented these information in the discussion on the observed increase of fluorescence upon stretching in the main text.

2. The interpretation of the threshold in the fluorescence increase needs some reconsideration. For example, for the hydrogel with D1, D1 is stronger than X/X* crosslinking domain. Therefore, it is highly possible that under tensile stress, X/X* crosslinking domain will dissociate earlier than D1, leading to larger strain threshold. And due to the dissociation of X/X* crosslinking domain, it may take longer time for D1 to reform.

This is an interesting remark, there are no doubts that, under strain, some X/X* open earlier or concomitantly to the sacrificial duplexes. Yet, indeed for D2, which is weaker than X/X*, the strain threshold is lower. All hydrogels break around the same strain, hence with similar overall X/X* reorganization. Therefore we believe that the effects of X/X* reconfiguration are secondary in the discussed difference of temporal recovery. It is the module architecture that controls the temporal recovery, and the partially mismatching, yet strong D1/D1 and D1*/D1* homoduplex are primarily responsible for the longer recovery time of D1 functionalized hydrogels. We now mention this discussion in the main text.

Reviewer #2 (Remarks to the Author):

The manuscript describes modular design of programmable mechanofluorescent DNA hydrogels. Sensitivity and modularity of mechanosensitive materials are known to be difficult to tune and/or to achieve. The authors could successfully design DNA modules to programmable hydrogels. The design has been precisely described, and the results are elegant. Several experimental supports are enough to admire the achievements. Figures 3 and 4 strongly support the concept. There are no points to be revised, and thus, the reviewer recommends the manuscript to accept without further revisions.

We thank the reviewer for his/her support.

Given the editor's comments on this review and the nature of the comments by reviewer 3, we would be glad if reviewer 2 could further strengthen his/her review regarding the progress and novelty this article brings to the field of mechanosensitive materials.

Reviewer #3 (Remarks to the Author):

The manuscript entitled "Modular Design of Programmable Mechanofluorescent DNA Hydrogels" should be rejected based on the following reasons.

We need to respectfully disagree with a range of the reviewer's comments and partly unjustified opinion of our manuscript, as it does not reflect the challenges regarding the design and characterization of macroscopic, mechanically responsive and sensing materials, as well as the broad impact that mechanochromic fluorescent hydrogels could have for soft materials design and also for their understanding. We try here to illustrate the frame and relevance of our research, specifically from a material perspective.

1. There is really nothing new in terms of technical logical innovation. Every single aspect of the technical designs is not new.

We agree that molecular force probes or DNA hydrogels or mechanofluorescent materials are known individually, yet the proposed approach combining macroscopic tensile and rheological characterization of sequence-controlled 3D DNA hydrogels and force probe-based mechanofluorescence is new and opens broad implications in the material field or to study cells in 3D. Furthermore, the described control of recovery (between reversible and irreversible) and examples of visualization of freezing strain patterns are completely new features on a mechanistic mechanical level and on a materials level. We now added additional citations of original works additionally to review on these techniques.

We want to stress that the implementation of molecular force probes into macroscopic 3D materials and their mechanical characterization is a key challenge in material design. While force-induced spiropyrene opening was known in solution (Potisek, S. L., Davis, D. A., Sottos, N. R., White, S. R., &

Moore, J. S. (2007) "Mechanophore-Linked Addition Polymers" *Journal of the American Chemical Society*, 129(45), 13808–13809.) their use to prepare macroscale mechanochromic glassy polymer made a great impact in the material field (Davis, D. A., Hamilton, A., Yang, J., Cremer, L. D., Van Gough, D., Potisek, S. L., et. Al. (2009) "Force-induced activation of covalent bonds in mechanoresponsive polymeric materials" *Nature*, 459, 68.).

Most importantly, as a general comment, even though design elements may be known to specific fields, their combination into new systems can lead to truly new behaviour and open routes to new conceptual design and new applications as shown by us in Figure 4 to shed new light onto materials behaviour.

2. The significance of the research outcome, in this case, the development of a fluorescent DNA Hydrogel sensitive to stretch, is unclear. The couple of examples shown in this study are unconvincing. In other words, there is no practical utility of this gel.

We disagree here with the reviewer. The reviewer does also not give any justification for why these examples are unconvincing. Macroscopic self-reporting materials that allow spatially resolved readout of molecular transformation are of key interest in the material field : (Sagara, Y. et al. Rotaxanes as Mechanochromic Fluorescent Force Transducers in Polymers. *J. Am. Chem. Soc.* 140, 1584-1587, (2018) ; Lavallo, P., Boulmedais, F., Schaaf, P. & Jierry, L. Soft-Mechanochemistry: Mechanochemistry Inspired by Nature. *Langmuir* 32, 7265-7276, (2016) ; Calvino, C., Neumann, L., Weder, C. & Schrettl, S. Approaches to polymeric mechanochromic materials. *J. Polym. Sci. A* 55, 640-652, (2017).).

The approach proposed here sets a new path based on precision molecular force probes to prepare 3D materials, and we show in Figure 4 clear examples on how these materials could be used for material characterization.

3. A major problem of this gel is that the color change is not correlated to physical distance displacement from the original position. As a result, talking about "programmable" is unfounded. There is nothing programmable at all. The outcome is not quantitative.

Our perspectives diverge here. As outlined in the manuscript and supplementary information the Change of Red/Green fluorescence is indeed quantitatively and reproducibly correlated to the strain applied. The frame of this study is the implementation of molecular force probes in macroscale materials and not the study of individual force process, hence, the absolute physical distance of each fluorophore from each quencher is irrelevant here as we observe averaged effects over many modules. In our non-perfect hydrogels, the abrupt opening observed for single tension probes (Zhang, Y., Ge, C., Zhu, C., & Salaita, K. (2014). DNA-based digital tension probes reveal integrin forces during early cell adhesion. *Nature Communications*, 5, 5167.) changes to a progressive increase of fluorescence due to the increasing density of open force-sensing modules. The R/G ratio is quantitatively correlated to macroscopic strain. As discussed above, the initial threshold distance corresponds to conformation rearrangements across the network that have lower energy than the bonds holding the duplex together.

Our system is programmable since we can decide between reversible and irreversible force sensing by molecular design depending on the sequence and architecture of the force sensing module. This property alone is rare enough among mechanochromic materials to be highlighted. Furthermore given the broad diversity of modules, fluorophore, DNA sequences and architectures that can be implemented in such system we are confident that the term programmable is fully justified.

4. The sensitivity of the mechanofluorescent DNA hydrogels is at best low. As shown in Figure 3, the distance change due to stretching must be huge before a color change can be seen or clearly detected. By then, however, the gel is broken. There is never a quantitative relationship between the color change and distance displacement. Hence, what is the use of this gel?

This is exactly why designing macroscale mechanochromic soft material is difficult. For macroscopic materials it is not the absolute distance between fluorophore/quencher that matters but the overall response measured macroscopically. Individual force probes, whose behaviour at the single molecule level or in confined interface (cell/cell, cell/surface...) may be relatively straightforward to identify, yield complex behaviours in macroscopic systems where multiple molecular reorganisation are taking place simultaneously. Here, the integration of a reference green fluorophore is a key feature enabling the quantitative strain/color correlation. These quantitative (and reproducible) measurements enabled us for example to identify that modules D1, HP and T present identical onset of fluorescence while onset for D2 is lower. Actually such a relatively large threshold was also observed for other soft mechanochromic materials (e.g Sagara, Y. et al. Rotaxanes as Mechanochromic Fluorescent Force Transducers in Polymers. *J. Am. Chem. Soc.* 140, 1584-1587, (2018)).

We would like to stress again that this work presents the first example of macroscopic mechanofluorescent materials based on DNA. There is no doubt that their sensitivity can be improved, using for example holliday junctions (Nickels, P. C., Wünsch, B., Holzmeister, P., Bae, W., Kneer, L. M., Grohmann, D., et.al. (2016). Molecular force spectroscopy with a DNA origami-based nanoscopic force clamp. *Science*, 354(6310)).

Yet, this work introduces the key conceptual and technical concept to further develop such macroscopic DNA hydrogel for various application across the biologic or material science fields.

5. The rheological study of these gels is grossly inadequate. For instance, the frequency sweep experiment should have been done, and temp should be changed as well. GN should have been done as well.

We now provide frequency sweep, amplitude sweep and temperature-dependant rheological characterization for the DNA hydrogel before and after functionalization (Supplementary Figure S5). We are confident that rheology experiments provided here show the key information required to understand properties and thermal behaviour of the DNA networks studied. Further rheological experiments and their interpretation would fit better in a separate article dedicated to a deeper understanding of the rheological properties of such DNA hydrogels.

6. Finally this reviewer has a lot of issues with the improper use of terminology and misleading terms or descriptions. Below are just some glaring examples.

We assume that misunderstandings concerning terminology originate from our different fields. We come from the physical chemistry and polymeric material fields and many of the terms used across our article are broadly accepted in this field and are not misleading. We tried to provide definitions and examples of use across the literature, and need to emphasize that the other reviewers have not pointed to such issues.

(1). The title is misleading. The fluorophore is not mechanosensitive at all. Further, the fluorescent intensity is not programmable. The authors have not done enough to show this can be programmable or “tunable” (as in the last paragraph on page 3).

It is the hydrogel as a whole that is mechanofluorescent – as stated in the title (“mechanofluorescent DNA hydrogel”). It could of course be reworded to mechanochromic fluorescent hydrogel as recently

published here (Sagara, Y. et al. Rotaxanes as Mechanochromic Fluorescent Force Transducers in Polymers. *J. Am. Chem. Soc.* **140**, 1584-1587, (2018).), but this does not make the title easier to understand and also there is in our opinion nothing incorrect with the terminology mechanofluorescent hydrogel.

As discussed above the fluorescence recovery of the hydrogel is indeed programmable/tuneable, and the module design offers many adjustable parameters.

(2). The authors do not do FRET measurements at all; yet they stated so in the Abstract. A color change is not a FRET-based detection at all.

We did not state that we perform FRET measurements i.e. on single molecule level. The detection of the strain is however based on a FRET mechanism at the core of the mechanical fluorescence activation. FRET (Förster Resonance Energy Transfer) refers to the mechanism describing energy transfer between two chromophores. Removing the acceptor chromophore, through mechanical strain, influences the FRET from the donor to the acceptor. Whether the acceptor fluoresces or not is irrelevant. Here, the increase of the distance increases the fluorescence of the donor by decrease of FRET. A broadly accepted definition of FRET is as follows "A donor chromophore, initially in its electronic excited state, may transfer energy to an acceptor chromophore through nonradiative coupling. The efficiency of this energy transfer is inversely proportional to the sixth power of the distance between donor and acceptor, making FRET extremely sensitive to small changes in distance." (Wikipedia or Lakowicz, J. R. *Principles of fluorescence spectroscopy*. (Springer Science, 2010))

It is indeed a FRET mechanism that is responsible for the change of fluorescence upon stretching, as fluorescence-enabled monitoring of strain is based on removal of FRET to the quencher molecule.

(3) The use of "mechano" as the prefix to describe the gel is misleading. The gels containing the "force sensor modules" can only respond to tension. The authors have not shown any other type of forces that can cause mechanical movement or displacement.

The prefix « mechano- » is broadly used for tension-sensitive system across the material field. We do not see the point why other mechanical forces, such as compression, need to be shown:

Some representative examples for the use of this terminology are as follows:

Fratzl, P. & Barth, F. G. Biomaterial systems for mechanosensing and actuation. *Nature* **462**, 442-448, (2009).

Davis, D. A. et al. Force-induced activation of covalent bonds in mechanoresponsive polymeric materials. *Nature* **459**, 68-72, (2009).

Lavalle, P., Boulmedais, F., Schaaf, P. & Jierry, L. Soft-Mechanochemistry: Mechanochemistry Inspired by Nature. *Langmuir* **32**, 7265-7276, (2016).

Sagara, Y. et al. Rotaxanes as Mechanochromic Fluorescent Force Transducers in Polymers. *J. Am. Chem. Soc.* **140**, 1584-1587, (2018).

(4) Fig 1 is fine to illustrate the idea. But there are lots of problems in this figure. (a) The use of "crosslinks" is troubling. Where is the cross-linker? (b) The nature of the bond that governs DNA base-pairing is hydrogen bond, not sacrificial bonds or duplex. It is just wrong. (c) The authors cannot rule out a heterogeneous formulation of the so-called "bar codes" between precursors A and B. Perhaps C and D may be involved as well. This may be true especially when precursor concentrations becomes

higher. The so-called mechanofluorescent module concentration could further contribute to the complexity of the polymer network structures. Yet, the authors have neither done a detailed study nor acknowledged these problems. These sources of heterogeneity could seriously affect the mechanical stability of the gels.

(a) Crosslinks are not necessarily covalent. Here the X/X* duplex forms multiple duplexes between the hydrogel precursors A and B across the materials which holds the hydrogel together, i.e. serving as supramolecular crosslinks.

(b) We are fully aware that in DNA, hydrogen bonds (as well as pi-stacking and the entropic release of water) yield the cooperative formation of a supramolecular helix, also referred to as DNA duplex. Macroscopically it is the sum of these hydrogen bonds holding the two DNA strands in a duplex which govern the mechanical properties of the hydrogel. The concept of sacrificial bond is broadly used across the field of biological materials (spider silk, bone etc.) and polymer mechanics as it plays an important role in toughness mechanisms. It does NOT state the nature of the bond but rather its behaviour upon mechanical actuation. Sacrificial bonds break before the overall failure of the material and dissipate energy. (Ducrot, E., Chen, Y., Bulters, M., Sijbesma, R. P. & Creton, C. Toughening Elastomers with Sacrificial Bonds and Watching Them Break. *Science* 344, 186-189, (2014); cited as ref. 24 in the original manuscript) The increase of fluorescence before failure of the material indicates that what we call « sacrificial duplex » indeed fails before the rupture of the hydrogel. Sacrificial bond is the correct term for the behaviour.

(c) We do not fully understand this sentence, there are no sequences called C or D in the RCA material, i.e. hydrogel precursors. There might be some interaction between the different domains of the hydrogel precursors (called A₁₋₄ and B₁₋₄). Yet, since we observe the melting (gel/sol transition) of the DNA hydrogel at T=T_m of X/X*, and because there is no significant property change for the pure hydrogels below this temperature, we are confident that X/X* primarily govern the stability of the hydrogel without the force-sensing modules. Note that we also checked the sequences using NUPACK to exclude any unwanted binding events/crosstalk. We now also provide rheological characterization after functionalization with D1 in Supplementary Figure S5. This experiment indeed shows relatively complex behaviour with temperature after the functionalization. Yet we want to stress that since we do not heat the functionalized hydrogels before mechanical testing we minimize any reorganizations and kinetically control the structure. In our protocol, the hydrogel functionalization takes place overnight, at low temperature and using already folded modules, hence we ensure that the modules remain properly folded and bind homogeneously in the hydrogel (as it is also visible from the homogeneous distribution of red or green fluorescence on all fluorescence microscopy images). Finally the hydrogels presented in this article sustain over 400 % of strain, which is well above that of most synthetic hydrogels, therefore we believe the concerns about the mechanical stability of the hydrogels are unjustified.

REVIEWERS' COMMENTS:

Editorial Note: Reviewer #1 was asked to comment on the authors' response to their own comments and to those of Reviewer #3.

Reviewer #1 (Remarks to the Author):

This reviewer think the authors have suitably addressed all reviewer concerns.

Reviewer #2 (Remarks to the Author):

The manuscript has been revised correctly, which seems to elevate its quality. No further revisions would be required.